# Extreme Hyperferritinemia: Causes and Prognosis

**DOI:** 10.3390/jcm11185438

**Published:** 2022-09-16

**Authors:** Maxime Fauter, Sabine Mainbourg, Thomas El Jammal, Arthur Guerber, Sabine Zaepfel, Thomas Henry, Mathieu Gerfaud-Valentin, Pascal Sève, Yvan Jamilloux

**Affiliations:** 1Department of Internal Medicine, University Hospital Croix-Rousse, Lyon 1 University, 69004 Lyon, France; 2CIRI, Centre International de Recherche en Infectiologie, Inserm U1111, Lyon 1 University, CNRS, UMR5308, ENS de Lyon, 69007 Lyon, France; 3Department of Internal Medicine and Vascular Pathology, University Hospital Lyon Sud, Lyon 1 University, 69000 Lyon, France; 4Laboratoire de Biométrie et Biologie Evolutive, Université de Lyon, Université Lyon 1, CNRS, UMR 5558, 69622 Villeurbanne, France; 5LIFE, Lyon Immunology Federation, 69000 Lyon, France; 6Department of Biochemistry and Molecular Biology, University Hospital Croix-Rousse, Hospices Civils de Lyon, 69004 Lyon, France; 7Research on Healthcare Performance (RESHAPE), INSERM U1290, Lyon 1 University, 69373 Lyon, France

**Keywords:** hyperferritinemia, hemophagocytic lymphohistiocytosis, septic shock, hyperferritinemic syndrome, ferritin

## Abstract

The significance of extreme hyperferritinemia and its association with certain diagnoses and prognoses are not well characterized. We performed a retrospective analysis of adult patients with at least one total serum ferritin (TSF) measurement ≥ 5000 µg/L over 2 years, in three university hospitals. Conditions associated with hyperferritinemia were collected, and patients were classified into 10 etiological groups. Intensive care unit (ICU) transfer and mortality rates were recorded. A total of 495 patients were identified, of which 56% had a TSF level between 5000 and 10,000 µg/L. There were multiple underlying causes in 81% of the patients. The most common causes were infections (38%), hemophagocytic lymphohistiocytosis (HLH, 18%), and acute hepatitis (14%). For TSF levels > 10,000 µg/L, there were no solid cancer or hematological malignancy without another cause of hyperferritinemia. Isolated iron-overload syndromes never exceeded TSF levels > 15,000 µg/L. Extreme hyperferritinemia (TSF levels > 25,000 µg/L) was associated with only four causes: HLH, infections, acute hepatitis and cytokine release syndromes. A total of 32% of patients were transferred to an ICU, and 28% died. Both ICU transfer rate and mortality were statistically associated with ferritin levels. An optimized threshold of 13,405 μg/L was the best predictor for the diagnosis of HLH, with a sensitivity of 76.4% and a specificity of 79.3%. Hyperferritinemia reflects a variety of conditions, but only four causes are associated with extreme hyperferritinemia, in which HLH and acute hepatitis are the most common. Extreme hyperferritinemia has a poor prognosis with increased mortality.

## 1. Introduction

Hyperferritinemia is usually defined by a level of total serum ferritin (TSF) exceeding 200 µg/L in women and 300 µg/L in men [1]. There are many causes of hyperferritinemia, including alcohol intake, liver disease, infection, cancer, chronic inflammation, or metabolic syndrome [2]. There is no consensual cut-off for extreme hyperferritinemia but, in previous studies, thresholds ranging from 2000 to 10,000 µg/L have been considered [3,4,5]. Extreme hyperferritinemia has mostly been associated with a limited number of causes, including septic shock, flares of Still’s disease, and hemophagocytic syndromes [3,6,7]. For some authors, these conditions could even belong to the same spectrum of diseases, grouped under the name of hyperferritinemic syndromes [8]. In this paradigmatic view, the dual role of ferritin (either immunosuppressive or pro-inflammatory) is the cornerstone of the pathogeneses of these conditions. However, the exact role hyperferritinemia plays in these diseases is still unclear.

Ferritinemia has also been studied as a prognostic biomarker [3,9]. Studies have shown a linear correlation between increased ferritin levels and all-cause mortality, with the magnitude of serum ferritin levels associated with a worse prognosis. Bennett et al. showed that very high ferritin levels were associated with more frequent intensive care unit (ICU) admissions and an increased mortality in children, whereas other authors have identified high ferritin as an independent risk factor for death in adult ICU patients [10,11].

The main objective of this study was to determine the main causes of hyperferritinemia and to analyze the associated prognosis in an unselected adult population, with a focus on extreme hyperferritinemia.

## 2. Materials and Methods

### 2.1. Data Collection

Using the electronic biological data management system (GLIMS 9–MIPS), we searched for all patients aged ≥18 years in three university hospitals (Hospices Civils de Lyon, Lyon, France), who had at least one TSF measurement between 1 January 2018 and 31 December 2019. For patients with multiple TSF measurements, the highest TSF value was selected for further analyses. Patient data were extracted automatically from an electronic patient data management system (EASILY-HCL). The follow-up period was defined as the time between the TSF measurement and the date of the last available data in the patient’s file (death or date of last visit). Epidemiologic data, medical causes, and outcomes were collected, as well as clinical features (body mass index, highest temperature, hepatomegaly, splenomegaly, and lymphadenopathy) and routine laboratory parameters (transferrin saturation, C-reactive protein (CRP), procalcitonin, aminotransferases, bilirubin, lactate dehydrogenase (LDH), serum creatinine, triglycerides, prothrombin, fibrinogen, and blood count).

### 2.2. Selection of TSF Cut-Offs

As we aimed to analyze the significance and prognosis of very high ferritinemia, the study focused on major and extreme hyperferritinemia. The previously used thresholds of 2000 and 10,000 µg/L yielded inconsistent results [3,7,12]. However, the 2000-µg/L threshold is too low and is not selective, and also results in too many underlying conditions [12]. On the contrary, the threshold of 10,000 µg/L is too high and selects only a limited number of causes. Therefore, an intermediate threshold of 5000 µg/L was retained because, in previous studies, this value corresponded to a balanced distribution of the ten major etiological groups (see below). For extreme hyperferritinemia, the cut-off point of 25,000 µg/L was chosen so as not to overlap with the 10,000 µg/L cut-off point used in some previous studies for major hyperferritinemia, and because this cut-off seemed clinically relevant.

### 2.3. Classification of Underlying Causes and Definitions

The literature review revealed ten major etiological groups classically analyzed in the setting of hyperferritinemia [4,6,13]. We retained these ten groups to yield the interpretation of our results in comparison with previous studies. The ten groups were: hemophagocytic lymphohistiocytosis (HLH), cytokine release syndrome (CRS), infections, rheumatic/inflammatory diseases, acute hepatitis, malignancy (either solid or hematological), iron overload, hemolysis, and a final group comprising all other rarer conditions. The classification of the underlying causes was based on the diagnoses made by the patient managing clinician (i.e., compatible clinico-biological presentation, disease course, treatments, and management). Two investigators (MF and YJ) independently reviewed the available data and confirmed the final diagnoses. Conflicting cases were discussed with a third expert (PS) and finally classified by consensus. The same protocol was used to determine the most likely underlying cause of hyperferritinemia (termed primary cause). The other underlying causes (termed secondary causes) were also recorded. 

The HLH-2004 criteria, the H-score and the 2016 EULAR/ACR/PRINTO criteria were used as the basis for the diagnosis of HLH [14,15,16,17]. HLH was always considered to be the primary cause of hyperferritinemia. The causes of HLH were classified into five subgroups: (1) infectious diseases; (2) hematological malignancies; (3) solid cancers; (4) rheumatic/inflammatory diseases; and (5) other conditions, including those of undetermined origin [18]. The diagnosis of virus-associated HLH was retained in case of a consistent clinical presentation and concomitant positive serum PCR. The diagnosis of an infectious disease was retained if the patient had a consistent disease course (i.e., severe infection requiring hospitalization or antimicrobials), even in the absence of a positive microbiological test. Infectious diseases were classified into five groups: bacterial, mycobacterial, viral, fungal, and parasitic. 

The diagnosis of rheumatic or inflammatory disease was made if the patient had an identified rheumatic disease in their history (treated or untreated), or if internationally recognized criteria were met [19,20,21,22,23,24,25].

The diagnosis of malignancy considered the clinical course, histological evidence when available, and/or exposure to anti-cancer therapy within the six months prior to testing. Malignancies were further subdivided into hematological malignancies and solid cancers. 

Cytokine release syndrome was only considered in the context of chimeric antigen receptor (CAR) T-cell therapy [26]. 

Acute hepatitis was defined by alanine or aspartate aminotransferase levels 10 times above the upper limit of normal, associated with identifiable acute liver disease [27].

Hemolysis was defined by anemia plus the biological stigmata of hemolysis (low haptoglobin, elevated LDH), typically in the context of autoimmune hemolytic anemia or hemoglobinopathies.

Iron-overload diagnosis was retained in patients with proven genetic hemochromatosis, those receiving regular red blood cell transfusions (i.e., ≥1/month for ≥6 months), or those requiring chelation therapy, with transferrin saturation > 45% [28].

Patients were considered immunocompromised if: (1) they had undergone stem cell or solid organ transplantation; (2) they had a solid or hematological malignancy; (3) they had received chemotherapy within the six months prior to the ferritin assay; (4) they were HIV infected; or (5) they were taking long-term immunosuppressive therapy (including steroids).

In this retrospective study, it was not possible to obtain data to calculate a severity score, such as the SOFA. Severity was therefore assessed by the need for transfer or intensive care management.

Outcomes were classified as: (1) ICU transfer, during the same hospitalization as the ferritin assay; (2) death within 6 months of the ferritin assay; or (3) favorable (neither ICU transfer nor death). The time between the ferritin assay and the transfer to an ICU or death was recorded.

### 2.4. Statistical Analyses

Baseline characteristics were assessed by descriptive statistics. Normally distributed continuous variables were described by their means and standard deviations, while continuous variables that are not normally distributed were described by their medians and ranges. Analysis of variance was performed for multiple comparisons of means. Categorical variables were described as numbers and percentages. They were compared using the Fisher’s exact test. Survival time was calculated from the date of the ferritin dosage to the date of censoring, the time of death, or the date of transfer to an ICU. Overall survival was estimated using the Kaplan–Meier method. Survival curves were compared using the log-rank test. To assess the ability of ferritin to serve as a biomarker of mortality, and to investigate the diagnostic efficacy of ferritin levels to diagnose HLH, a receiver operating characteristic (ROC) curve was used, and the area under the curve (AUC) of the ROC curve was calculated. The cut-off value of ferritin levels was determined by the maximum Youden’s index. For mortality prediction, the optimal cutpoint of mortality risk was determined from the maximally selected rank statistic using the survminer, maxstat, and optimal cutpoints package. The multivariate Cox regression approach was used to identify independent variables correlated with mortality, and the results were presented as hazard ratios (HR) with 95% confidence intervals (CI). For all statistical analyses, *p* < 0.05 was considered significant. Analyses were performed using R (R Foundation for Statistical Computing, Vienna, Austria).

## 3. Results

### 3.1. Patient Characteristics 

During the 2-year inclusion period, there were 142,051 TSF tests conducted. A total of 813 (0.57%) tests, corresponding to 495 patients, were over the cut-off of 5000 µg/L. The characteristics of the patients are shown in Table 1. The mean age was 56 (±17) years, and the men-to-women ratio was 1.3. Most patients (88%) were immunocompromised and/or had a history of cancer (78%), mostly hematological malignancies. The maximum TSF level was 1,133,280 µg/L. Among patients with TSF levels > 5000 µg/L, 56% had a TSF level between 5000 and 10,000 µg/L, 36% between 10,000 and 50,000 μg/L, and 8% had a TSF level > 50,000 µg/L (Appendix A). Most patients (*n* = 325, 66%) had two causes of hyperferritinemia, while 93 (19%) patients had a single underlying cause, and 77 (16%) patients had three. 

### 3.2. Etiological Distribution

The distribution of primary and secondary causes with the related median TSF level and inflammatory parameters are detailed in Table 2. Increasing numbers of underlying causes (one, two or three) per patient were significantly associated with increasing TSF levels (the respective means were 12,300 µg/L for one cause; 22,610 µg/L for two; and 57,241 µg/L for three, *p* > 0.005).

The most common primary underlying causes were infectious diseases (38%). Among the 114 infections with available microbiological evidence, bacterial infections were the most prevalent (66%), followed by fungal infections (22%) and viral infections (18%). Of note, some patients had several simultaneous infections.

HLH was the second most prevalent primary cause (*n* = 92, 19%), and was associated with another cause of hyperferritinemia in 98% of cases. HLH was associated with hematological malignancies in 78% of cases, infections in 9%, and solid cancers or rheumatic/inflammatory diseases in 7%. There was one patient with primary HLH (NRLC4-MAS). 

Acute hepatitis (*n* = 74, 15%) was also frequently associated with another cause of hyperferritinemia (solid cancer, 20%; infection, 19%; hematological malignancy, 12%; rheumatic/inflammatory disease, 10%). The remaining cases were mostly drug-induced (15%), alcoholic (11%), or secondary to heart failure (5%).

Sixty-six (13%) patients had iron-overload syndrome, which was always transfusion-induced (i.e., iron overload in patients with malignancies or hemoglobinopathies).

Likewise, patients with active malignancies, without an added cause (e.g., HLH, infection, hepatitis), rarely had a TSF level > 5000 µg/L (*n* = 32, 6%). 

Twenty-three (5%) patients had underlying rheumatic/inflammatory disease. In nine of them, a disease flare was the primary cause of hyperferritinemia, while in the other fourteen, hyperferritinemia was due to a complication: either HLH, hepatitis, or infection.

Fifteen (3%) patients had CRS following CAR T-cell therapy for refractory lymphoma (*n* = 13) or acute leukemia (*n* = 2), and seven (1.5%) patients had acute hemolysis (in the context of hemoglobinopathy or autoimmunity). Nine patients could not be classified into one of the ten previous groups. 

The distribution of the primary causes of hyperferritinemia in the groups stratified by TSF level is shown in Figure 1. Interestingly, a TSF level > 10,000 µg/L was never associated with an isolated solid or hematological malignancy (i.e., without another cause of hyperferritinemia) and a TSF level > 15,000 µg/L was never associated with post-transfusion iron overload. As TSF levels increased, the proportion of infectious causes gradually decreased, while that of HLH gradually increased. The proportion of acute hepatitis was evenly distributed between the TSF groups. 

Finally, only four diagnoses were associated with extreme hyperferritinemia (*n* = 95, 19%): half the cases were due to HLH, and the remainder were due to CRS, acute hepatitis, and infectious diseases (mean TSF, 126,200 µg/L; 120,711 µg/L 55,255 µg/L and 43,407 µg/L, respectively).

### 3.3. Outcomes and Prognosis

A total of 43% of patients with TSF levels > 5000 µg/L had an unfavorable outcome, of which 32% (*n* = 160) required ICU transfer and 28% died (*n* = 141). At 6 months, the overall mortality was 47%. In patients with TSF levels > 25,000 µg/L, the 6-month mortality raised to 61%. Both ICU transfer rate and mortality were statistically associated with TSF levels (*p* < 0.001, Figure 2). Furthermore, multivariate analysis showed that TSF levels were significantly associated with mortality, independent of age, sex, and underlying causes (*p* < 0.001, Appendix A). Specifically, each 10,000 µg/L increase in TSF concentration was associated with a 3% increase in mortality per day (HR: 1.03, 95% CI: 1.02–1.04, *p* < 0.001). Other inflammatory parameters were not influenced by TSF levels: for example, there was no significant difference in CRP and WBC between patients with TSF levels ≥ 25,000 and <25,000 µg/L (137 and 134, *p* = 0.42; and 8.4 and 10.7, *p* = 0.12, respectively). 

The prognostic value of TSF levels was then investigated. TSF levels > 14,660 µg/L were associated with a maximized mortality risk (46% vs. 20% during hospitalization, *p* < 0.001). The diagnosis of underlying HLH was significantly associated with increased mortality at 6 months (54% vs. 46%, *p* < 0.001). Independent of TSF levels, the diagnosis of HLH significantly increased the risk of death compared to other causes (HR: 1.6, 95% CI: 1.07–2.40, *p* = 0.022). Within the HLH group, the TSF level was also predictive of death and a TSF level > 19,450 µg/L was associated with higher mortality (*p* < 0.001).

### 3.4. Performance of TSF for the Identification of HLH 

Discriminating factors between patients with HLH and patients with other causes were then sought (Table 3). In patients with TSF levels ≥ 5000 μg/L, the mean TSF level was significantly higher in patients with HLH than in all other disease groups (*p* < 0.001). 

The ROC curve analysis revealed an optimized threshold of TSF levels at 13,405 μg/L, giving a sensitivity (Se) of 76.4% and a specificity (Sp) of 79.3% for the diagnosis of HLH (AUC = 0.824; *p* < 0.001). As expected, the mean H-score was significantly higher in patients with HLH (212 vs. 125, *p* < 0.001). In addition, most components of the H-score were significantly different in the HLH group (i.e., ASAT, triglycerides, fibrinogen, cytopenia, maximal temperature, and organomegaly). We tested the previously reported H-score cut-off of 169 [15] and found a Se of 81%, a Sp of 81%, a positive predictive value (PPV) of 50%, and a negative predictive value (NPV) of 95% for HLH prediction. The ROC curve analysis revealed that, in our cohort of patients with TSF levels ≥ 5000 µg/L, the best H-score threshold for the detection of HLH was 160, giving a Se of 90% and a Sp of 78%, a PPV of 48%, and a NPV of 98% (AUC = 0.900; *p* < 0.001). Increased LDH, which was not included in the H-score, was significantly associated with the diagnosis of HLH (*p* < 0.001). The optimized LDH threshold for the detection of HLH was 953 U/L (Se, 77%; Sp, 57%). Mean CRP and fibrinogen levels were significantly different between HLH and non-HLH patients (169 vs 128 mg/L and 3.5 vs. 4.8 g/L, respectively, *p* < 0.001), and they were especially low in patients with acute hepatitis (mean, 81 mg/L and 2.7 g/L). Hemophagocytosis, investigated in 46/92 patients with HLH, was present in 31 (67%) cases. 

## 4. Discussion

In this study, we have shown that hyperferritinemia is associated with a wide spectrum of conditions. However, three causes accounted for more than two-thirds of cases of major hyperferritinemia (i.e., TSF levels ≥ 5000 µg/L): infectious diseases, HLH, and acute hepatitis. Furthermore, when TSF levels reached extreme values (i.e., >25,000 µg/L), only the diagnoses of HLH, infections, acute hepatitis, and CRS remained. 

Extreme hyperferritinemia has classically been associated with a limited number of conditions and a poor prognosis, regardless of the underlying cause [29]. In the literature, the etiological spectrum of extreme hyperferritinemia varies greatly between studies. While similar conditions are described (i.e., malignancy, iron overload syndromes, infections, HLH, rheumatic/inflammatory diseases, hemolysis, and liver or kidney diseases), there is a large diversity in their frequencies [5,6,7,30,31]. For example, in 111 patients with TSF levels > 10,000 µg/L, Senjo et al. showed a higher frequency of infections and liver diseases [12], while Otrock et al. reported a predominance of malignancies in 583 patients [31], and Sackett et al. reported a higher prevalence of iron-overload syndromes in 269 patients [4]. These differences may be related to the different study populations and recruitment methods. In the absence of a consensual threshold for extreme hyperferritinemia, previous studies are difficult to compare as they selected different thresholds, most often between 1000 and 10,000 µg/L [3,5,7,13,32]. For instance, Belfeki et al. defined extreme hyperferritinemia as TSF levels > 2000 µg/L whereas the threshold was 10,000 µg/L in the study by Sackett et al. [3,4]. 

Consistent with other studies, we have shown that infectious diseases are the most common underlying causes of major hyperferritinemia [7,12,30]. Under these inflammatory conditions, ferritin levels are influenced by the cytokine environment and ferritin is mobilized, without excess total iron in the body, to act as an iron chelator against bacterial invasion [33]. Because the severity of infections was rarely assessed in previous studies, it is impossible to distinguish between sepsis and septic shock.

We also showed that: (i) isolated iron overload syndromes never led to TSF levels > 15,000 µg/L; and (ii) isolated malignancies never led to TSF levels > 10,000 µg/L. 

Hematological malignancies accounted for 3% of major hyperferritinemia when it was the only underlying cause. However, 56% of patients with either HLH, acute hepatitis, or infection had an associated hematological malignancy. In the literature, the frequency of hematological malignancies ranges from 2 to 38% of patients, depending on its interpretation as the primary or secondary cause of hyperferritinemia, while solid malignancies remain very rare [6,12,30]. 

Our study is one of the few to include patients with CRS, probably because CAR T-cell therapy is a recent therapeutic option. Like interleukin-6, hyperferritinemia has been repeatedly reported as a good biomarker of CRS following CAR T-cell therapy [34]. CRS was one of the four causes of extreme hyperferritinemia (mean, 120,711 µg/L). CAR-T-cell-associated CRS is almost never a difficult diagnosis because it occurs in a particular context.

Interestingly, while catastrophic antiphospholipid syndrome has been previously associated with extreme hyperferritinemia [35], we did not identify any cases in our cohort. A larger series may be needed to better estimate the frequency of this condition, which remains exceptional [36]. 

Extreme hyperferritinemia was associated with poor prognosis, as well as an ICU transfer requirement and mortality, which were both statistically associated with serum ferritin levels. In intensive care medicine, the predictive performance of ferritin as an outcome marker has been reported [10,11,33]. Previous studies have shown that decreasing TSF levels during treatment are predictive of a favorable outcome in patients with sepsis [37]. In contrast, increasing levels are predictive of a poor outcome in patients hospitalized with influenza [38]. Ferritin was also reported as an independent factor of increased mortality, regardless of the underlying cause [3,9]. Finally, in line with previous studies, we have shown that, regardless of its cause, HLH is associated with increased mortality, with ferritinemia being an independent predictor of mortality [3,11,39].

While extreme hyperferritinemia is highly suggestive of HLH, it is not a specific biomarker [5,7,31]. In our cohort, half of the patients with TSF levels > 25,000 µg/L had underlying HLH. In the literature, the frequency of HLH causing hyperferritinemia is reported to range from 4 to 17% [3,5,30,32]. Belfeki et al. observed an HLH frequency of 14% in patients with TSF levels ≥ 2000 µg/L, but a frequency of 62% in patients with TSF levels ≥ 6000 µg/L [3]. Schram et al. reported a frequency of 17% in a population with TSF levels ≥ 50,000 µg/L [7]. In our study, such a cut-off would have resulted in a frequency of 67.5%, suggesting that other determinants prevented comparison between studies. For example, the high percentage of hematological malignancies (56%) in our study may be, at least partly, responsible for this difference. 

We found that a TSF threshold of 13,406 μg/L was the best predictor of HLH. Previous studies have reported lower values, ranging from 2000 to 16,000 µg/L [11,30,40]. In ICU patients, Saeed et al. reported an optimal cut-off at 3951 µg/L (Se, 88% and Sp, 82%) [40], whereas this cut-off was 9083 μg/L (Se, 92.5% and Sp, 91.9%) in Lachmann’s study [11] and 16,000 µg/L (Se, 79.4 and Sp 79.2) in Naymagon’s study [30]. The latter was identified in hospitalized patients with TSF levels ≥ 5000 µg/L [30].

Finally, we found that elevated LDH levels were significantly associated with HLH, suggesting this parameter could be added to the H-score criteria to enhance its performance [15].

Our study has limitations, the main one being its retrospective nature, which limited the availability of data (especially the severity assessment). In addition, although a non-selected population, our cohort was drawn from tertiary medical centers, with specialized recruitment that may have affected the etiological distribution. Variations in the timing of ferritin measurements may have affected ferritin values.

## 5. Conclusions

Extreme hyperferritinemia is suggestive of four underlying causes, with HLH and infections being the most common. Although not specific for HLH, extreme values should raise the possibility of this diagnosis. Major and extreme hyperferritinemia have a poor prognosis and increasing ferritin levels are associated with increasing mortality.

## Figures and Tables

**Figure 1 jcm-11-05438-f001:**
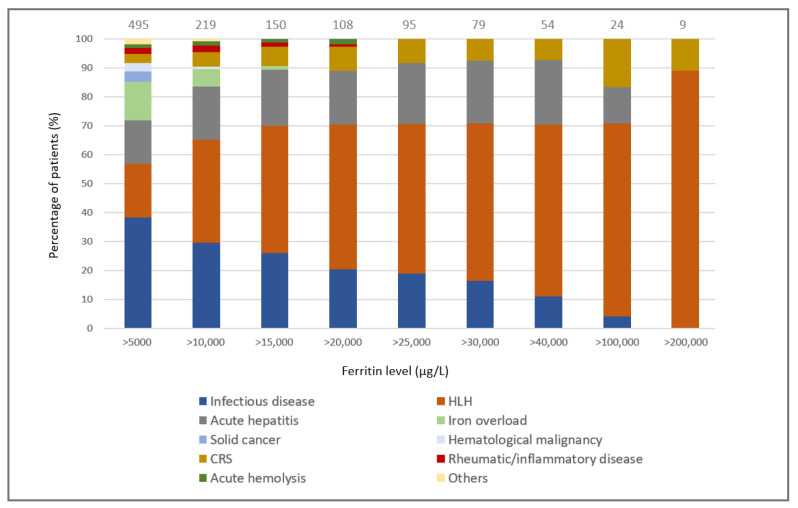
Distribution of the primary causes of hyperferritinemia according to total serum ferritin levels. The number of the patients in each group is indicated.

**Figure 2 jcm-11-05438-f002:**
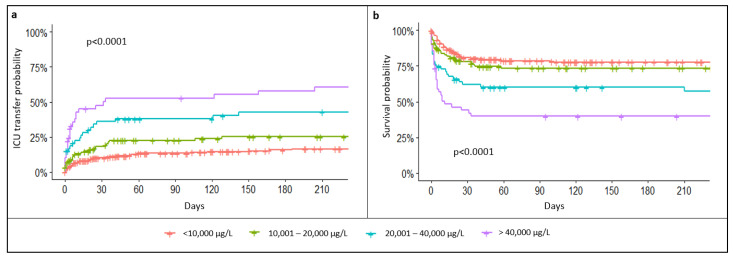
Probability of ICU transfer (**a**) and survival probability (**b**) according to total serum ferritin levels.

**Table 1 jcm-11-05438-t001:** Characteristics of patients with ferritinemia > 5000 μg/L.

Patient Characteristics	
Patients, *n* (%)	495 (100)
Age, mean (SD), years	56.6 (17.2)
Gender, *n* (%)	
Male Female	281 (56.8) 214 (43.2)
Median follow-up, days (range)	129 (1–1149)
Median serum ferritin level, µg/L (range)	9128 (5013-1,133,280)
Median serum C-reactive protein, mg/L (range)	104 (1–614)
Immunosuppression, *n* (%)	436 (88)
History of cancer, *n* (%)	388 (78)
Number of causes by patient, *n* (%)	
1 cause 2 causes or more	93 (18.8) 402 (81.2)

**Table 2 jcm-11-05438-t002:** Primary causes of hyperferritinemia ≥ 5000 µg/L with related median ferritin, CRP and WBC.

Causes	N (%)	Median Ferritin, µg/L (IQR)	Median CRP, mg/L (IQR)	Median WBC, (G/L-IQR)
**Infectious diseases**	190 (38.4)			
Not documented	76 (15.3)	8176 (6217–12,537)	170 (65–255)	5.5 (1.2–12.2)
Bacterial	75 (15.1)			
Fungal	25 (5.0)			
Viral	20 (4.0)			
Mycobacterial	3 (0.6)			
Parasitic	2 (0.4)			
**Hemophagocytic lymphohistiocytosis**	92 (18.6)			
Hematological malignancy	68 (13.7)	27,392 (14,400–59,506)	159 (61–250)	2.9 (1.2–6.9)
Infections	29 (5.9)			
* Viral*	*21 (4.2)*			
* Bacterial*	*3 (0.6)*			
* Fungal*	*2 (0.4)*			
* Mycobacterial*	*2 (0.4)*			
Solid cancer	7 (1.4)			
Rheumatic/inflammatory diseases	6 (1.2)			
* AOSD*	*2 (0.4)*			
* DRESS*	*2 (0.4)*			
* SLE*	*2 (0.4)*			
Primary HLH	1 (0.2)			
* NLRC4-MAS*	*1 (0.2)*			
Drug-induced	1 (0.2)			
Idiopathic	2 (0.4)			
**Acute hepatitis**	74 (14.9)			
Solid cancer	15 (3.0)	11,366 (6760–25,648)	37 (15–115)	7.5 (4.4–12.7)
Infections	14 (2.8)			
* Viral*	*5 (1.0)*			
* Bacterial*	*4 (0.8)*			
* Fungal*	*2 (0.4)*			
Hematological malignancy	9 (1.8)			
Drug-induced	11 (2.2)			
Alcoholic	8 (1.6)			
Rheumatic/inflammatory diseases	7 (1.4)			
Others	10 (2.0)			
**Iron-overload syndromes (post-transfusion)**	66 (13.3)			
In patients with malignancies	54 (10.9)	6342 (5653–9191)	19 (9–55)	7.5 (3.7–11.9)
In patients with hemoglobinopathy	12 (2.4)			
**Malignancies**	32 (6.4)			
Solid cancer	17 (3.4)	6312 (5630–7738)	186 (34–251)	8.2 (5.7–16.4)
Hematological malignancy	15 (3.0)			
**CRS**	15 (3.0)	27,368 (8980–79,558)	25 (16–168)	0.9 (0.3–2.2)
**Rheumatic/inflammatory diseases**	10 (2.0)			
AOSD	3 (0.6)	10,052 (6016–12,441)	159 (99–361)	11.2 (4.3–15.7)
DRESS	2 (0.4)			
Sweet Syndrome	2 (0.4)			
Castleman disease	1 (0.2)			
TAFRO syndrome	1 (0.2)			
Polymyalgia rheumatica	1 (0.2)			
**Acute hemolysis**	7 (1.4)			
Hemoglobinopathy	3 (0.6)	6457 (6062–17,314)	22 (15–119)	15.3 (9.4–17.4)
Autoimmune hemolytic anemia	2 (0.4)			
Thrombotic microangiopathy	2 (0.4)			
**Unclassified**	9 (1.8)	9066 (6206–9975)	75 (9–122)	7.7 (6.4–8.1)

Primary causes are in bold font. Abbreviations: AOSD: adult-onset Still’s disease; CRP: C-reactive protein; DRESS: drug rash with eosinophilia and systemic symptoms; IQR: interquartile range; NLRC4-AID: NRLC4-associated autoinflammatory disease; SLE: systemic lupus erythematosus; TAFRO: thrombocytopenia, anasarca, fever, reticulin fibrosis, organomegaly. Unclassified: primary causes that could not be included in the other groups; WBC: white blood count.

**Table 3 jcm-11-05438-t003:** Comparison of clinical and laboratory characteristics of the study population according to different etiological groups.

	HLH	Infections	Hepatitis	R/I Diseases *	Others	*p* Value
**Patients (N)**	92	190	74	10	129	
**Age (mean, years)**	55.6	58.0	54.2	48.4	57.4	0.23
**Ferritin (mean, µg/L)**	74,607	12,410	22,055	11,019	15,062	**<0.001**
**CRP (mean, mg/L)**	168	172	81	217	84	**<0.001**
**PCT (mean, µg/L)**	16.9	21.2	3.6	12.0	1.0	0.502
**ASAT (mean, U/L)**	1340	253	2000	78	123	**<0.001**
**ALAT (mean, U/L)**	381	131	1258	87	99	**<0.001**
**LDH (mean, U/L)**	2579	848	1849	428	857	**<0.001**
**Creatinin (mean, µmol/L)**	98	113	99	119	100	0.613
**Triglycerides (mean, mmol/L)**	3.5	2.4	2.4	2.7	2.5	**0.001**
**Fibrinogen (mean, g/L)**	3.5	5.7	2.7	6.2	4.5	**<0.001**
**Hemoglobin (mean, g/L)**	86.8	84.9	107.3	95.6	90.8	**<0.001**
**Platelets (mean, G/L)**	64	106	165	181	149	**<0.001**
**Leukocytes (mean, G/L)**	5.0	11.6	10.6	11.4	11.9	**0.03**
**Neutrophils (mean, G/L)**	3.5	7.9	8.4	9.9	7.2	**0.04**
**Lymphocytes (mean, G/L)**	1.1	1.9	1.1	0.9	1.8	0.352
**BMI (mean)**	24.6	23.5	23.9	24.9	23.8	0.444
**Temperature (mean, °C)**	39.2	38.1	37.5	39.0	37.1	**<0.001**
**Hepatomegaly (% of patients)**	0.22	0.06	0.32	0.10	0.11	**<0.001**
**Splenomegaly (% of patients)**	0.25	0.04	0.08	0.20	0.10	**<0.001**
**Lymphadenopathy (% of patients)**	0.38	0.10	0.04	0.40	0.15	**<0.001**
**H-score (mean)**	212	136	123	147	109	**<0.001**

* R/I diseases: Rheumatic/inflammatory diseases.

## Data Availability

Not applicable.

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
