# Peer review of "Extreme Hyperferritinemia: Causes and Prognosis"

_jcm, 2022, doi:10.3390/jcm11185438_

Round 1

Reviewer 1 Report

Manuscript ID jcm-1913027 "Extreme hyperferritinemia: causes and prognosis”

 A study that evaluates the causes of extreme hyperferritinemia and the factors affecting mortality in detail. I have a few suggestions:

Castleman disease and DRESS syndrome are not primary rheumatologic diseases. It should not be included in the rheumatological disease group or the group name should be replaced as rheumatological/inflammatory disease??? or the alternative by some other way.

Detailed demographic and laboratory findings of the groups are given in Supplementary Table 2. This table contains important information. This table should be in the main text, not in the Supplementary section.

Author Response

- Castleman disease and DRESS syndrome are not primary rheumatologic diseases. It should not be included in the rheumatological disease group or the group name should be replaced as rheumatological/inflammatory disease??? or the alternative by some other way.

=> the group name has been modified to rheumatic / inflammatory diseases

- Detailed demographic and laboratory findings of the groups are given in Supplementary Table 2. This table contains important information. This table should be in the main text, not in the Supplementary section.

=> the table has been added to the main text

Reviewer 2 Report

This is a retrospective analysis of adult patients with at least one total serum ferritin (TSF) measurement ≥5,000 μg/L over 2 years, in three university hospitals.Extreme hyperferritinemia (TSF>25,000 μg/L) were associated with only four causes: HLH, infections, acute hepatitis, and cytokine release syndromes. 32% of patients were transferred to an ICU, and 28% did both ICU transfer rate and mortality were statistically associated with ferritin levels and extreme hyperferritinemia has a poor prognosis with increased mortality. This study is innovative, and the study design is fair. Hyperferritinemia is an indicator of metabolic disarrangement, and it is not surprising to see a close correlation between infection and hyperferritinemia.

The author should provide some other indicators of infection/inflammation(WBC, CPR) measured along with the ferritin to provide the concomitant infection/inflammation

status when they detected hyperferritinemia. This will provide more solid evidence between hyperferritinemia and the extent of infection/inflammation and also the association between hyperferritinemia 

patient prognosis.

Author Response

Thank you for your comments which will improve the manuscript.

Inflammatory parameters have been included in the manuscript, especially in the Table 2 with addition of CRP and WBC, as well in the population description.
Comparison of CRP and WBC between different TSF levels has also been added :
"Other inflammatory parameters were not influenced by TSF levels, with, for example, no significant difference in CRP and WBC between patients with TSF ≥25,000 and < 25,000 µg/L (respectively 137 and 134, p=0.42; and 8.4 and 10.7, p=0.12)."